# Craniofacial Morphologic Predictors for Passive Myofunctional Therapy of Pediatric Obstructive Sleep Apnea Using an Oral Appliance with a Tongue Bead

**DOI:** 10.3390/children9071073

**Published:** 2022-07-18

**Authors:** Yi-Jing Hwang, Yu-Shu Huang, Yun-Chia Lian, Yu-Hsuan Lee, Michele Hervy-Auboiron, Chung-Hsing Li, Cheng-Hui Lin, Li-Chuan Chuang

**Affiliations:** 1Department of Pediatric Dentistry, Chang Gung Memorial Hospital at Linkou, Taoyuan City 333, Taiwan; msgiliboom@cgmh.org.tw (Y.-J.H.); yunyun445@cgmh.org.tw (Y.-C.L.); celine0608@hotmail.com (Y.-H.L.); 2Department of Child Psychiatry and Sleep Center, Chang Gung Memorial Hospital, Chang Gung University, Taoyuan City 333, Taiwan; yushuhuang1212@gmail.com; 3Graduate Institute of Craniofacial and Dental Science, College of Medicine, Chang Gung University, Taoyuan City 333, Taiwan; 4Orthodontic Institute, 93130 Noisy-Lesec, France; michelehervy@me.com; 5Division of Orthodontics and Dentofacial Orthopedics and Pediatric Dentistry, Department of Dentistry, Tri-Service General Hospital, Taipei City 100, Taiwan; chiyenchli@yahoo.com.tw; 6Craniofacial Center and Craniofacial Research Center, Chang Gung Memorial Hospital, Chang Gung University, Taoyuan City 333, Taiwan; clementlin0614@gmail.com; 7Department of Dentistry, School of Dentistry, National Yang-Ming University, Taipei City 112, Taiwan

**Keywords:** obstructive sleep apnea, children, oral appliance, craniofacial predictor, morphologic predictor

## Abstract

We conducted this retrospective study to identify potential clinical, polysomnographic, and cephalometric predictors for the treatment outcomes of a tongue-beaded oral appliance (OA) in children with obstructive sleep apnea syndrome (OSAS). In total, 63 patients—50 boys and 13 girls ranging in age from 4 to 16 years—underwent OA treatment nightly for at least 6 months. A baseline digital lateral cephalometric radiograph was obtained for each patient. Multivariate logistic regression analysis was performed to examine predictors for the treatment outcome based on the clinical and cephalometric measurements. Overall, 28 patients responded to the treatment (post-treatment improvement > 50% or apnea–hypopnea index (AHI) < 1/h), and 35 did not (post-treatment improvement < 50% and AHI ≥ 1/h). Significantly larger cranial base angle (SNBa), smaller lower gonial angle (LGo Angle), and shorter length of anterior cranial base (SN) were found in responders. Smaller lower gonial angle (LGo Angle) and smaller anterior cranial base (SN) predict a favorable outcome for pediatric OSAS using a tongue-beaded OA. This finding will equip practitioners with additional insights when selecting suitable candidates for OA therapy in pediatric patients.

## 1. Introduction

Obstructive sleep apnea syndrome (OSAS), a common sleep-disordered-breathing (SDB) condition, is characterized by repeated cessations of breathing during sleep with a partial or complete airway obstruction. It could occur in children of all ages, from neonates to adolescents [1,2,3]. Various clinical consequences may also be triggered by OSAS, such as daytime sleepiness, mouth breathing, behavioral difficulties, nocturnal enuresis, drooling, and snoring during sleep. OSAS could even lead to severe complications, if left untreated. 

Pediatric OSAS could adversely affect children’s development and general health, and increase metabolic, cardiovascular, and neurocognitive morbidities. Heightened emotional disturbances and inattention during the daytime have also been reported [4,5,6,7]. Hence, early treatment is strongly recommended to mitigate or eliminate these negative consequences.

OSAS in children is defined as an AHI (apnea–hypopnea index) ≥ 1 event per hour based on the diagnosis of polysomnography (PSG), a criterion adopted by the American Academy of Sleep Medicine. Reported prevalence rates of pediatric OSAS ranged from 1% to 4%, depending on the age of the study subjects, method of assessment, and selection criteria of the study [8,9,10,11,12,13]. The risk factors commonly associated with OSAS are either anatomical or neuromuscular, including craniofacial anomalies, adenotonsillar hypertrophy, obesity, upper airway inflammatory process, environmental exposure, asthma, prematurity, and genetics [9,13,14].

Adenotonsillectomy (T&A) has been the first-line treatment for children with adenotonsillar hypertrophy and severe OSA [2,3]. For mild to moderate OSA or residual persistent obstruction, adjunctive therapies may be beneficial, for instance, nasal continuous positive airway pressure (nCPAP), intranasal corticosteroid, weight loss, orthodontic treatment, oral appliance (OA), mandibular advancement device (MAD), and myofunctional therapy (MFT) [3,5,15,16,17].

OA is an important alternative to nCPAP for treating mild OSA and residual post-operative symptoms. It is better tolerated than nCPAP due to its quietness and portability, as well as the comfort it provides [18,19,20,21]. These oral devices hold the mandible in a protrusive and increased vertical position during sleep, thereby preventing the pharyngeal obstruction by retraining the muscle tone of the tongue and further affecting the pharyngeal structure [22,23]. Studies have shown that the use of OAs effectively reduced OSA and related symptoms in adult populations [19,24,25,26]. Two randomized controlled studies also demonstrated the effectiveness of MADs for pediatric OSAS [27,28].

Several studies have investigated the cephalometric characteristics of craniofacial and airway morphology in children with OSAS. Most of these children exhibited retrusive mandible, large overjet, long face appearance, and divergent mandibular growth patterns [29].

Since 1995, there have been a number of studies aiming to identify predictors for the treatment outcome in adults with different types of OA [26,30,31,32,33,34,35,36,37]. However, similar investigations have not been conducted in children with OSAS to date. Therefore, the purpose of this retrospective study is to identify potential clinical, polysomnographic, and cephalometric factors that may predict the treatment outcome of an OA with a built-in tongue bead in children with OSAS. The insights derived from this investigation will help the selection of suitable candidates who are more likely to benefit from this treatment.

## 2. Materials and Methods

The Institutional Review Board of the Human Investigation Committee of the Chang Gung Memorial Hospital and the Chang Gung University approved this study (IRB104-9308A3). Written informed consent was read and signed by each participant and their legal guardian. Patients also signed informed consent regarding publishing their data and photographs.

### 2.1. Participants

The inclusion criteria included: (1) age ranged from 4 to 16, (2) diagnosis of OSA (AHI ≥ 1/h) based on overnight PSG at the Sleep Center of the Chang Gung Memorial Hospital in Linkou (Taoyuan, Taiwan), (3) having been wearing an oral appliance overnight for at least 6 months, and (4) availability of cephalometric radiographs. The exclusion criteria included: epilepsy, head injury, severe developmental delay and mental retardation, schizophrenia, severe depression, and inability to cooperate with overnight PSG. Patients with severe hypertrophic tonsil or adenoid tissues were also excluded.

The medical records of children with OSA who were treated at the Chang Gung Memorial Hospital from March 2014 through January 2020 were reviewed. A total of 63 children—50 boys and 13 girls—met the inclusion criteria. Their mean age was 8.75 ± 3.24 years. The distribution of grades of OSA was as follows: 42 (66.7%) with mild OSA, 16 (25.4%) with moderate OSA, and 5 (7.9%) with severe OSA (Table 1).

Post-treatment improvement was calculated using the following formula. Patients were classified into two groups based on the post-treatment outcome: the responsive group (post-treatment improvement > 50% or AHI < 1/h), and the non-responsive group (post-treatment improvement < 50% and AHI ≥ 1/h).
$$\mathrm{Post}\text{-}\mathrm{treatment}\;\mathrm{improvement}=\frac{\left(\mathrm{pre}\text{-}\mathrm{treatment}\;\mathrm{AHI}-\mathrm{post}\text{-}\mathrm{treatment}\;\mathrm{AHI}\right)}{\mathrm{pret}\text{-}\mathrm{reatment}\;\mathrm{AHI}}\times 100\%$$

### 2.2. Lateral Cephalometric Radiography

A baseline digital lateral cephalometric radiograph was obtained for each patient. Patients were instructed to sit in an upright position and keep their heads in the natural position, with the Frankfort horizontal plane parallel to the floor. Patients were directed to breathe at ease with their teeth in centric occlusion and lips closed in a relaxed position. All digital lateral cephalometric radiographs were traced by the same investigator (Y.-J. Hwang) using a digital image software package (GE Medical Systems Inc. Released 2006. Centricity Enterprise, Version 3.0., USA: GE Healthcare Inc., Chicago, IL, USA), and verified by an experienced pedodontist (L.-C. Chuang). A total of 24 cephalometric measurements were taken for each patient: 9 angular, and 15 linear. All landmarks, variables, and their definitions are detailed in Figure 1 and Table 2.

To assess the error rate, 20 randomly selected radiographs were measured and traced again under the same condition and at least 2 weeks after the initial measure. According to the Dahlberg formula, the mean (SD) of the error rate was 0.6 (0.5) mm (range: 0.0–1.9 mm) for linear variables and 0.5 (0.4)° (range: 0.0–1.8°) for angular variables. The following clinical variables were also recorded prior to the OA treatment: sex, age, birth weight, gestational age, full-term or preterm delivery, body mass index (BMI), OSA severity, and PSG data.

### 2.3. Oral Appliance with a Built-In Tongue Bead

All patients underwent OA treatment nightly for more than half a year. The single-piece OA used for this study includes a built-in tongue bead and a custom-made acrylic plate for the upper arch. The tongue bead is mounted on the lower end of the frame for the tip of the tongue to roll. The wearer’s mandible is thus placed in a forward and downward position with the tongue in a forward position to open the airway [38,39].

An interocclusal record was taken with the mandible at half of the maximum jaw protrusion and a minimum vertical separation of about 2 mm. Patients were instructed to put on the OA before bed and roll the bead with their tongue during sleep. The OA is a hybrid of MAD and passive MFT (European Patent No. 288,82,384, 12 October 2016; US Patent No. 10,105,2056, 23 October 2018).

Recall appointments were arranged for each patient every 3 months to check the condition and fitting of the oral device, as well as any side effect or discomfort from wearing the device. The device would be fixed or adjusted, if needed.

## 3. Statistical Analyses

Data were computed and analyzed using a laptop and a statistical software package (IBM SPSS Statistics for Windows, version 20.0, IBM Corp., Armonk, NY, USA). Means and standard deviations for all variables were calculated for each group. Chi-square analysis was conducted for cross-group comparison, including sex, preterm or full-term delivery, and severity of OSA. The Mann–Whitney U test was carried out for other baseline and cephalometric measurements. Intragroup comparisons of pre- and post-treatment polysomnographic measures for the responsive and non-responsive group, respectively, were performed using Wilcoxon sign-rank test. Statistical significance was set at *p* value less than 0.05 for all analyses.

A multivariate logistic regression analysis was performed to examine predictors for the treatment outcome based on the clinical and cephalometric measures. A receiver operating characteristic (ROC) curve was also drawn to measure sensitivity, specificity, positive predictive value (PPV), and negative predictive value (NPV) of the model.

## 4. Results

Out of the 63 cases reviewed, 28 were classified as responsive (44.4%), and 35 cases, as non-responsive (55.6%) (Table 1).

### 4.1. Descriptive Statistics

There were no statistically significant differences between the two groups regarding sex, age, birth weight, gestational age, BMI, severity of OSA, and baseline PSG data (Table 1). Significant post-treatment improvement was seen in AHI, RDI, mean SpO_2_ (%), average SpO_2_ (%), and minimum SpO_2_ (%) in the responsive group (Table 3). Contrariwise, the post-treatment AHI was higher than the AHI before the treatment in the non-responsive group (Table 4).

### 4.2. Cephalometric Analysis

Statistically significant differences were associated with the following cephalometric measures between the responsive and the non-responsive groups: cranial base angle (SNBa) (132.22 degrees ± 4.32 vs. 130.27 degrees ± 4.31; *p* = 0.041), lower gonial angle (LGo Angle) (77.65 degrees ± 3.33 vs. 79.38 degrees ± 2.95; *p* = 0.017), and anterior cranial base (SN) (63.84 mm ± 3.53 vs. 66.15 mm ± 4.31; *p* = 0.029) (Table 5). Responders’ LGo Angle and SN were significantly smaller than those of the non-responders, and responders’ cranial base angle (SNBa) was significantly larger than that of non-responders. No statistically significant differences were seen between the two groups for measurements related to the airway space, hyoid bone position, or dental relationship.

Multivariate logistic regression analysis results showed that LGo Angle and SN were important predictors for the treatment outcome (Table 6). The prediction equation had moderate sensitivity (67.9%), moderate specificity (77.1%), moderate PPV (70.4%), and moderate NPV (75%). The probability (P) of treatment success was calculated as follows:$$P\;\left(\mathrm{OA}\;\mathrm{treatment}\;\mathrm{success}\right)=\frac{\exp\left(27.309-0.210\times \mathrm{LGo}\;\mathrm{Angle}-0.171\times \mathrm{SN}\right)}{1+\exp\left(27.309-0.210\times \mathrm{LGo}\;\mathrm{Angle}-0.171\times \mathrm{SN}\right)}$$

As shown in the ROC curve in Figure 2, the regression model accurately predicted 73% of total cases (responders and non-responders), indicating an acceptable discrimination threshold.

Some minor side effects and complaints were reported, including excessive salivation, as well as mild jaw muscle and tooth discomfort the following morning. These conditions generally improved over time.

## 5. Discussion

To the best of our knowledge, the current study was the first to explore predictors for the treatment outcome of an oral device in pediatric patients with OSAS. Previous studies have focused on adult patients, and the definitions of being responsive to treatment also varied. Nonetheless, all previous studies concluded that treating OSAS with OAs was effective and that certain craniofacial structures did impact the OA treatment outcome.

Our study found that, after the OSA treatment using an oral device with a built-in tongue bead, the craniofacial measures of SNBa, LGo Angle, and SN differed significantly between responders and non-responders. The significantly shorter SN measure might be related to the age and shorter body height of these young patients, suggesting a more pronounced positive effect of such intervention in younger patients. Among responders, the anterior facial height (NMe) was also smaller than among non-responders, although the difference was not statistically significant. There could be a link between this finding and responders’ stronger predilection for skeletal class II [40] and hypodivergent mandibular growth tendency.

Shen et al. [37] reported a favorable OA treatment outcome in patients with a shorter anterior facial height (AFH). Eveloff et al. [30] presented a prediction equation where posterior facial height (PFH) correlated positively with the post-treatment AHI associated with a removable Herbst appliance. Our findings were consistent with these observations.

In the meantime, not all findings derived from our study paralleled those from earlier studies.

Investigators in previous studies have found that a larger SNA [30,31], smaller SNB [31,36,37], and larger intermaxillary discrepancy (ANB) [36] had a fair predictive ability for the treatment success of OAs. Endo et al., reported a larger SNB and a smaller ANB in the responsive group than in the non-responsive group [33]. Our study showed, instead, that, despite the correlation of a smaller SNB and larger ANB with the treatment success, the difference between responders and non-responders did not reach statistical significance. It is also worth noting that no studies have demonstrated that SNBa, LGo Angle, and SN were associated with a higher probability of successful OA treatment. Hence, the association emerging from our study between a larger SNBa, smaller SN, and smaller LGo Angle and positive OA treatment outcome appears novel.

Interestingly, there were no statistically significant differences in airway-related measurements between the two groups in our study, which echoed findings among the Japanese population [33]. This was contrary to reports from a number of studies that smaller superoposterior airway space (SPAS) [32], posterior airway space (PAS) [30,31], middle airway space (MAS) [31,35], and inferior airway space (IAS) [32] had a fair predictive ability for the effectiveness of OA therapy and post-treatment AHI. In addition, Shen et al. [37] observed that a smaller minimal retroglossal airway (minRGA) correlated with the success of the OA treatment. The oropharyngeal cross-sectional area (OROXA) was found to be smaller in responders than in non-responders [32,35].

Nevertheless, findings from these studies all pointed to a more favorable treatment outcome of OAs among OSAS patients with a narrower oropharyngeal airway space. Practitioners should thus exert caution when treating patients with a wider anteroposterior airway [35]. At the same time, retropalatal airway space (RPAS) has been found by Mehta et al., to negatively impact the post-treatment AHI, using a mandibular advancement splint [26]. Evidently, the clinical evidence presented in the literature remains inconclusive.

Despite the backward and superior position of the hyoid bone among responders in our study—more so than among non-responders, the difference between the two groups was not statistically significant. However, previous studies have found a smaller hyoid-mandibular plane distance (MPH) among responders [30,33]. This finding might suggest that the influence of the airway morphology on OSA could be the result of functional adaptation.

For AHI, studies have reported a more favorable treatment outcome of OA among patients with a lower baseline AHI [26,30,36]. Yet, the baseline AHI and pretreatment severity of OSA were comparable between the responsive and non-responsive groups in our study.

In terms of sex, age, overjet, and overbite, they did not correlate with the response to the oral device used in our study. Although responders in our study were younger than non-responders in age (years) and gestational age (weeks), the differences between the two groups were not statistically significant. There have been, however, reports of favorable responses to MADs among women than men [34,41]. Liu et al., observed that the younger the patient, the better the response to OAs [32]. Hoekema et al., also concluded that overjet and overbite had a fair predictive ability for the response to the OA therapy [36].

MADs improve the lower airway patency and increase the size of upper airway through mandibular advancement and vertical opening to alleviate or treat the condition of OSA [22]. The vertical opening between the upper and lower incisors results in a downward rotation of the mandible. Hence, caution needs to be exercised when treating patients with clockwise growth pattern—a hyperdivergent skeletal pattern—as the MADs may aggravate patients’ condition of OSA. In fact, unfavorable treatment outcomes associated with MADs have been seen by Shen et al. [37] among patients with long face appearance, a feature similar to the hyperdivergent skeletal pattern. Our study also found a significantly larger SNBa and smaller LGo Angle among treatment responders (than non-responders) who presented a hypodivergent skeletal or counterclockwise growth pattern.

Some limitations of our study should also be mentioned. First, sleep apnea is more likely to occur when patients are in a supine position [42]. Yet, the lateral cephalometric radiograph was taken when patients were sitting in an upright position and looking straight forward. Hence, more sophisticated techniques could have been employed to simulate the dynamic changes of the pharyngeal airway in order to more accurately predict the OA treatment outcome and identify influential neuromuscular factors.

Second, if the OA used in our study had been adjustable and could advance the mandible incrementally, we would have been able to better fine-tune the measurement. Given these young patients’ tendency to behave uncooperatively, it was difficult to set a precise value of mandibular advancement, and the difference in the maximum mandibular advancement does affect the treatment outcome [36]. In addition, our study results could not definitively distinguish the treatment effect between passive myofunctional therapy (PMFT) using a tongue-beaded device and the amount of mandibular advancement. Furthermore, the appliance may trigger stretch receptors, which, in turn, activate the airway supporting muscles and increase the airway patency [43].

Third, in terms of patients’ compliance with the treatment protocol, we relied mostly on the reporting of their legal guardians at recalls. To obtain more complete information to assist the evaluation of the treatment outcome, questionnaires could have been created to collect input from legal guardians on patient compliance, subjective symptoms, perceived treatment efficacy, side effects, and satisfaction.

Fourth, sleep posture has also been found to significantly impact the effectiveness of oral appliance therapy [42]. It was shown that the mean AHI dropped significantly when patients were in the supine position or the prone position. Hence, if patients’ sleep postures had been recorded, it would assist our investigation of the treatment outcome of the oral device used in our study.

Future research can recruit a larger sample used to confirm and expand the prediction equation proposed in the current study. It can also broaden the scope to examine long-term effects of MADs. Although polysomnography is a valuable tool to determine the improvement in OSA, few studies have utilized other outcome measures. For studies that did, the results remained inconclusive, and, thus, the absence of such measures from our study. Last but not least, to gain a more complete understanding of the clinical effectiveness of myofunctional therapy in children with OSA, future studies can examine other symptoms related to OSA, such as snoring, and daytime symptoms, such as sleepiness and hyperactivity.

## 6. Conclusions

Smaller lower gonial angle (LGo Angle) and smaller anterior cranial base (SN) predict a favorable outcome of an OA treatment using a tongue-beaded device for pediatric OSAS. This finding will equip practitioners with additional insights when selecting suitable candidates for OA therapy in pediatric patients.

## Figures and Tables

**Figure 1 children-09-01073-f001:**
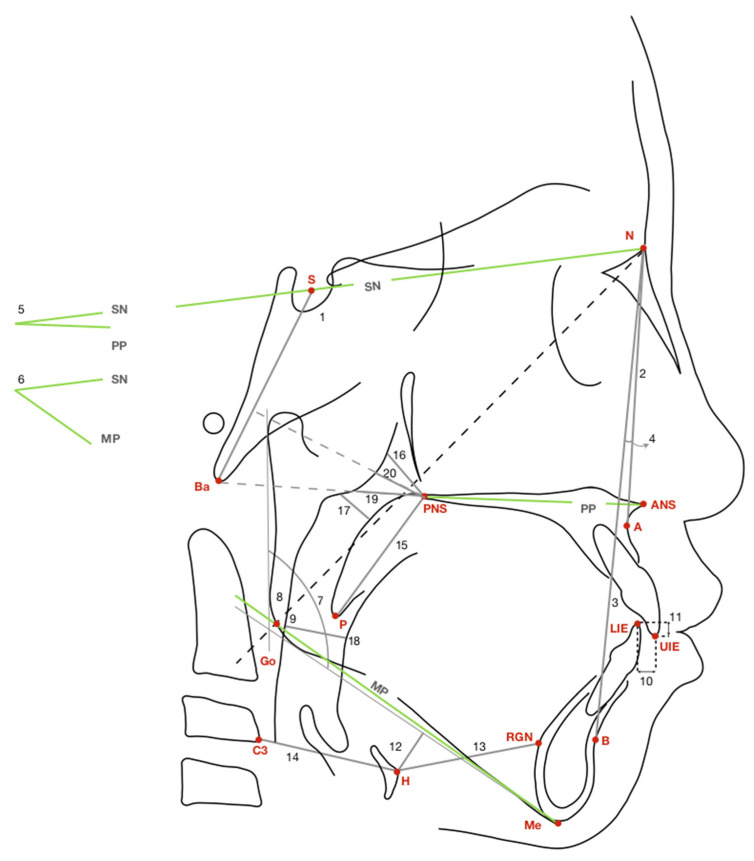
A diagrammatic representation of landmarks, reference lines, and variables used to identify the facial skeleton, teeth, upper airway, soft palate, tongue, and hyoid bone on a cephalometric radiograph. A: Subspinale. ANS: Anterior nasal spine. B: Supramentale. Ba: Basion. C3: Most anteroinferior point of third cervical vertebral body. Go: Gonion. H: Most anterosuperior point of hyoid bone. LIE: Lower incisor edge. Me: Menton, the most inferior point on symphyseal outline. MP: Mandibular plane, plane constructed from Me through Go. N: Nasion. P: Most inferior tip of soft palate. PNS: Posterior nasal spine. PP: Palatal plane, plane constructed from ANS through PNS. RGN: Retrognathion. S: Sella. SN: Line from sella to nasion. UIE: Upper incisor edge. 1. SNBa; 2. SNA; 3. SNB; 4. ANB; 5. SNPP; 6. SNMP; 7. Gonial angle; 8. Upper gonial angle; 9. Lower gonial angle; 10. Overjet; 11. Overbite; 12. MP-H; 13. H-RGN; 14. Hy-C3; 15. LSP; 16. PNSNPhp; 17. MinRPA; 18. MinRGA; 19. PNSAD1; and 20. PNSAD2.

**Figure 2 children-09-01073-f002:**
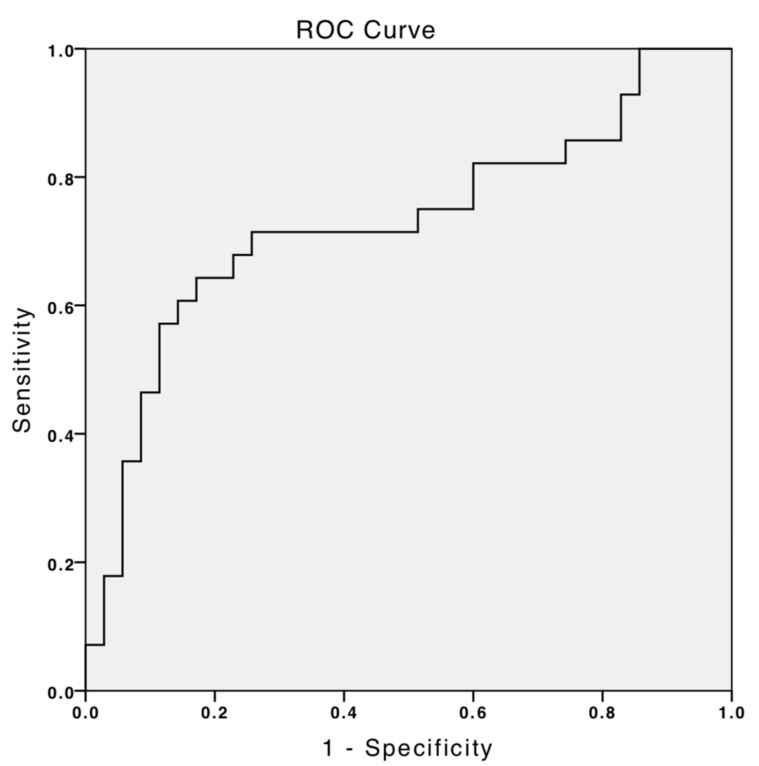
A receiver operating characteristic (ROC) curve was drawn based on the prediction equation for the treatment outcome of a tongue-beaded OA. The area under the ROC (AUC) equals 0.729, indicating an acceptable discrimination threshold.

**Table 1 children-09-01073-t001:** Descriptive statistics.

	Responsive Group (*N* = 28)	Non-Responsive Group (*N* = 35)	Total (*N* = 63)	*p*
Sex, no. (%)				
Male	21 (33.3%)	29 (46.1%)	50 (79.4%)	0.444 ^a^
Female	7 (11.1%)	6 (9.5%)	13 (20.6%)	
Age (years)	8.14 ± 3.45	9.23 ± 3.03	8.75 ± 3.24	0.140 ^b^
Birth weight (gm)	2692.32 ± 804.59	2931.4 ± 781.70	2825.14 ± 794.58	0.368 ^b^
Gestational age (week)	36.39 ± 3.77	37.77 ± 3.00	37.16 ± 3.40	0.119 ^b^
Birth, no. (%)				
Full-term	16 (25.4%)	25 (39.7%)	41 (65.1%)	0.237 ^a^
Preterm	12 (19%)	10 (15.9%)	22 (34.9%)	
BMI	17.90 ± 4.42	19.39 ± 5.18	18.72 ± 4.88	0.240 ^b^
Body height (cm)	125.68 ± 22.42	129.28 ± 29.28	127.68 ± 26.31	0.265 ^b^
Pre-treatment OSA severity, no. (%)				
Mild (1 ≤ AHI < 5)	17 (27.0%)	25 (39.7%)	42 (66.7%)	0.546 ^a^
Moderate (5 ≤ AHI < 10)	9 (14.3%)	7 (11.1%)	16 (25.4%)	
Severe (AHI ≥ 10)	2 (3.2%)	3 (4.8%)	5 (7.9%)	
Pretreatment PSG data				
AHI (No./h)	4.79 ± 3.39	4.41 ± 4.03	4.58 ± 3.73	0.575 ^b^
RDI (No./h)	7.46 ± 5.89	6.59 ± 5.01	6.98 ± 5.39	0.534 ^b^
Mean SpO_2_ (%)	97.27 ± 0.87	97.57 ± 0.65	97.44 ± 0.76	0.088 ^b^
Average SpO_2_ (%)	96.46 ± 2.62	97.11 ± 0.83	96.83 ± 1.87	0.601 ^b^
Minimum SpO_2_ (%)	89.61 ± 4.25	89.94 ± 3.80	89.79 ± 3.97	0.994 ^b^
Post-treatment PSG data				
AHI (No./h)	1.42 ± 1.25	5.66 ± 4.99	3.78 ± 4.34	0.000 *^,b^
RDI (No./h)	4.47 ± 4.77	6.59 ± 5.34	5.65 ± 5.16	0.007 *^,b^
Mean SpO_2_ (%)	97.75 ± 0.52	97.51 ± 0.70	97.62 ± 0.63	0.156 ^b^
Average SpO_2_ (%)	97.46 ± 0.58	96.94 ± 7.65	97.17 ± 0.73	0.005 *^,b^
Minimum SpO_2_ (%)	92.18 ± 4.06	89.89 ± 4.25	90.90 ± 4.29	0.009 *^,b^

Responsive group: Participants with more than 50% improvement in post-treatment AHI or post-treatment AHI less than 1 event per hour. Non-responsive group: Participants with less than 50% improvement in post-treatment AHI and post-treatment AHI more than 1 event per hour. * *p* < 0.05, ^a^ Chi-square test, ^b^ Mann–Whitney U test. BMI: body mass index. PSG: polysomnography. AHI: apnea–hypopnea index. RDI: respiratory disturbance index. SpO_2_: oxygen saturation.

**Table 2 children-09-01073-t002:** Cephalometric analysis: variables and definitions.

**Skeletal Measurement, Degree**	
SNBa	Cranial base angulation in midsagittal plane
SNA	Angle from S to N to A point
SNB	Angle from S to N to B point
ANB	Anteroposterior discrepancy degree of maxilla and mandible
SNPP	Angulation of the palatal plane with the SN line
SNMP	Angulation of the mandibular plane with the SN line.
Gonial Angle (Go Angle)	The angle formed by the tangent to the posterior border and the tangent to the lower border of the mandible
Upper gonial angle (UGo Angle)	The superior angle of gonial angle divided by N-Go line
Lower gonial angle (LGo Angle)	The inferior angle of gonial angle divided by N-Go line
**Skeletal measurement, mm**	
SN	Anterior cranial base length, from sella to nasion
SBa	Length of sella to basion
SGo	Posterior facial height, from sella to gonion
NMe	Anterior facial height, from nasion to menton
**Dental measurement, mm**	
Overjet	Horizontal relation of maxillary and mandibular incisors
Overbite	Vertical relation of maxillary and mandibular incisors
**Hyoid bone measurement, mm**	
MP-H	Linear distance between hyoid bone (H) to mandibular plane.
H-RGN	Linear distance between *H* and *RGN*.
Hy-C3	Linear distance between *C3* and *H*.
**Airway measurement, mm**	
LSP	Soft palate length
PNSNPhp	Width between PNS and posterior wall of nasopharynx
MinRPA	Minimal width of airway behind soft palate perpendicular to posterior pharyngeal wall
MinRGA	Minimal width of airway behind tongue perpendicular to posterior pharyngeal wall
PNSAD1	Width from PNS to the nearest adenoid tissue measured along PNS-Ba line
PNSAD2	Width from PNS to the nearest adenoid tissue measured along the line perpendicular to S-Ba line

**Table 3 children-09-01073-t003:** Pre- and post-treatment polysomnographic measures in the responsive group.

	Pre-Treatment				Post-Treatment				*p*
	Mean	SD	Minimum	Maximum	Mean	SD	Minimum	Maximum	
AHI (No./h)	4.79	3.39	1.50	14.20	1.42	1.25	0.00	5.70	0.000 *
RDI (No./h)	7.46	5.89	2.10	28.90	4.47	4.77	0.80	21.70	0.002 *
Mean SpO_2_ (%)	97.27	0.87	94.00	98.00	97.75	0.52	96.00	98.00	0.007 *
Average SpO_2_ (%)	96.46	2.62	84.80	98.00	97.46	0.58	96.00	98.00	0.018 *
Lowest SpO_2_ (%)	89.61	4.25	79.00	95.00	92.18	4.06	78.00	97.00	0.000 *

* *p* < 0.05, Wilcoxon sign-rank test. SD: standard deviation. AHI: apnea–hypopnea index. RDI: respiratory disturbance index. SpO_2_: oxygen saturation.

**Table 4 children-09-01073-t004:** Pre- and post-treatment polysomnographic measures in the non-responsive group.

	Pre-Treatment				Post-Treatment				*p*
	Mean	SD	Minimum	Maximum	Mean	SD	Minimum	Maximum	
AHI (No./h)	4.41	4.03	1.10	20.90	5.66	4.99	1.40	22.90	0.035 *
RDI (No./h)	6.59	5.01	1.70	23.50	6.59	5.34	1.80	23.20	0.993
Mean SpO_2_ (%)	97.57	0.65	95.00	98.00	97.51	0.70	96.00	98.00	0.686
Average SpO_2_ (%)	97.11	0.83	95.00	98.00	96.94	7.65	95.00	98.00	0.275
Lowest SpO_2_ (%)	89.94	3.80	77.00	95.00	89.89	4.25	75.00	95.00	0.846

* *p* < 0.05, Wilcoxon sign-rank test. SD: standard deviation. AHI: apnea–hypopnea index. RDI: respiratory disturbance index. SpO_2_: oxygen saturation.

**Table 5 children-09-01073-t005:** Cephalometric analysis: responsive group vs. non-responsive group.

Measurement	Responsive Group (*N* = 28)	Non-Responsive Group (*N* = 35)	Total (*N* = 63)	*p*
	Mean ± SD	Mean ± SD	Mean ± SD	
Skeletal measurement				
SNBa (degree)	132.22 ± 4.32	130.27 ± 4.31	131.13 ± 4.39	0.041 *
SNA (degree)	80.49 ± 2.39	81.43 ± 3.73	81.01 ± 3.22	0.391
SNB (degree)	75.59 ± 2.50	76.97 ± 3.93	76.36 ± 3.42	0.139
ANB (degree)	4.80 ± 2.19	4.49 ± 2.45	4.62 ± 2.33	0.580
SNPP (degree)	8.56 ± 2.86	8.49 ± 2.93	8.52 ± 2.87	0.994
SNMP (degree)	37.87 ± 3.96	37.75 ± 4.32	37.80 ± 4.13	0.928
Go Angle (degree)	126.11 ± 5.41	128.98 ± 5.78	127.70 ± 5.75	0.058
UGo Angle (degree)	48.46 ± 4.15	49.59 ± 4.86	49.09 ± 4.56	0.290
LGo Angle (degree)	77.65 ± 3.33	79.38 ± 2.95	78.61 ± 3.22	0.017 *
SN (mm)	63.84 ± 3.53	66.15 ± 4.31	65.12 ± 4.12	0.029 *
SBa (mm)	43.82 ± 4.07	44.95 ± 4.68	44.45 ± 4.42	0.189
SGo (mm)	72.75 ± 8.82	75.74 ± 8.96	74.41 ± 8.95	0.116
NMe (mm)	113.55 ± 11.66	117.76 ± 10.49	115.89 ± 11.14	0.184
Dental measurement				
Overjet (mm)	3.64 ± 1.57	3.17 ± 2.31	3.38 ± 2.01	0.422
Overbite (mm)	2.75 ± 1.77	3.04 ± 2.11	2.91 ± 1.96	0.494
Hyoid b. measurement				
MPH (mm)	10.94 ± 5.76	12.76 ± 5.07	11.95 ± 5.42	0.118
HRGN (mm)	30.25 ± 7.08	30.94 ± 7.12	30.63 ± 7.06	0.552
HyC3 (mm)	32.24 ± 5.43	34.26 ± 5.12	33.36 ± 5.31	0.118
Airway measurement				
LSP (mm)	31.05 ± 4.27	32.00 ± 3.88	31.58 ± 4.05	0.251
PNSNPhp (mm)	15.41 ± 3.67	15.56 ± 3.72	15.49 ± 3.67	0.846
MinRPA (mm)	6.84 ± 2.83	7.44 ± 2.87	7.18 ± 2.85	0.430
MinRGA (mm)	11.87 ± 3.08	12.42 ± 3.52	12.17 ± 3.32	0.520
PNSAD1 (mm)	20.64 ± 4.07	20.66 ± 5.02	20.65 ± 4.59	0.750
PNSAD2 (mm)	15.98 ± 3.92	15.86 ± 3.76	15.91 ± 3.81	0.978

* *p* < 0.05.

**Table 6 children-09-01073-t006:** Multivariate logistic regression analysis: predictors for treatment outcome of a tongue-beaded oral appliance.

Variable	Beta *	Odds Ratio (95% Confidence Interval)	*p*-Value
LGo Angle, Degrees	−0.210	0.811 (0.678–0.970)	0.022
SN, Millimeters	−0.171	0.843 (0.733–0.970)	0.017

* Regression coefficient.

## Data Availability

The data supporting the findings of this study are private due to the protection of personal data.

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
