# Peer review of "Craniofacial Morphologic Predictors for Passive Myofunctional Therapy of Pediatric Obstructive Sleep Apnea Using an Oral Appliance with a Tongue Bead"

_children, 2022, doi:10.3390/children9071073_

Round 1

Reviewer 1 Report

Dear Authors,

The study is good and sheds the light on an important of Craniofacial morphologic predictors for passive myofunctional therapy of pediatric patient.

Proofreading for this manuscript to correct minor grammatical errors is recommended.

In the Discussion part:

the 1st paragraph needs to be modified.

Reviewer 2 Report

nicely done. simple presentation. stats seem fine, although im not an expert. 

Author Response

We appreciate very much Reviewer’s kind words and encouraging feedback.

Reviewer 3 Report

The discussion would benefit from clarity. No other significant issues noted

Author Response

Discussion has been revised to enhance clarity.

Reviewer 4 Report

Dear authors, the research is very interesting, but the weakest part is subjects selection. The age 4-16 years is to wide, and the patients are not in the same phase of growth and development. Also I suggest to monitor the patients longer than  6 months because they were looking the changes made by functional treatment, and to make comparison of changes after 6 months an after maybe 3 additional months.

I would also recommend  deleting the following sentences at the beginning of the chapter discussion: Authors should discuss the results and how they can be interpreted from the perspective of previous studies and of the working hypotheses. The findings and their implications should be discussed in the broadest context possible. Future research directions may also be highlighted

Round 2

Reviewer 4 Report

Dear authors , thank You for taking into the consideration my review.